# Lysosomes as a Target of Anticancer Therapy

**DOI:** 10.3390/ijms24032176

**Published:** 2023-01-22

**Authors:** Wojciech Trybus, Ewa Trybus, Teodora Król

**Affiliations:** Department of Medical Biology, Jan Kochanowski University in Kielce, 25-406 Kielce, Poland

**Keywords:** lysosomes, cathepsins, lysosomal membrane permeability, apoptosis, autophagy

## Abstract

Lysosomes are organelles containing acidic hydrolases that are responsible for lysosomal degradation and the maintenance of cellular homeostasis. They play an important role in autophagy, as well as in various cell death pathways, such as lysosomal and apoptotic death. Various agents, including drugs, can induce lysosomal membrane permeability, resulting in the translocation of acidic hydrolases into the cytoplasm, which promotes lysosomal-mediated death. This type of death may be of great importance in anti-cancer therapy, as both cancer cells with disturbed pathways leading to apoptosis and drug-resistant cells can undergo it. Important compounds that damage the lysosomal membrane include lysosomotropic compounds, antihistamines, immunosuppressants, DNA-damaging drugs, chemotherapeutics, photosensitizers and various plant compounds. An interesting approach in the treatment of cancer and the search for ways to overcome the chemoresistance of cancer cells may also be combining lysosomotropic compounds with targeted modulators of autophagy to induce cell death. These compounds may be an alternative in oncological treatment, and lysosomes may become a promising therapeutic target for many diseases, including cancer. Understanding the functional relationships between autophagy and apoptosis and the possibilities of their regulation, both in relation to normal and cancer cells, can be used to develop new and more effective anticancer therapies.

## 1. Introduction

Neoplastic diseases, despite significant advances in treatment methods, are still the second most common cause of death, right behind cardiovascular diseases. One of the main reasons for the failure of anticancer therapy is the phenomenon of multidrug resistance to the cytostatics used, often occurring in cancer cells as a result of various interactions [1]. The problem of multidrug resistance appears to be extremely important because it accounts for more than 90% of deaths in oncology patients treated with traditional chemotherapeutics or new targeted therapy drugs [2]. The main factors contributing to resistance to cancer treatment are, among others, inappropriate dosing, changes in the rate of drug entry into the cell, enhanced metabolism of xenobiotics, increased removal of drugs from the cell through the intensified expression of transport proteins, inactivation of drugs, increased ability to repair DNA and inhibition of apoptosis [2,3,4]. Loss of control over the process of apoptosis allows cancer cells to both survive and accumulate mutations that contribute to increased invasiveness, enhanced angiogenesis, impaired cell proliferation and differentiation [5]. In cancer cells, apoptosis is usually inhibited by the overexpression of anti-apoptotic proteins and a low expression of pro-apoptotic proteins, which may be the main cause of treatment failure when using chemotherapy.

Thus, the task of modern oncology is to search for new strategies to overcome the resistance of cancer cells to the therapies used, as well as to understand the mechanisms of this phenomenon.

Lysosomal cell death, which occurs with the participation of the lysosomal system, may play an important role in anticancer therapy. The main reason for drawing attention to this type of cell death is the specific structure and role of lysosomes in both normal and malignant transformed cells, which makes lysosomes a unique target for the action of anticancer drugs. This work focuses on the mechanism of lysosomal death, based on the destabilization of the lysosomal membrane and on pharmacologically interesting compounds whose potential can be used to induce lysosomal membrane permeability. Compared to other types of cell death, such as apoptosis, lysosomal cell death can be very useful in eliminating the phenomenon of multidrug resistance, which is one of the main causes of failure when using anticancer therapies.

## 2. Structure and Functions of Lysosomes

Lysosomes are organelles found in eukaryotic cells in all mammalian cells except mature erythrocytes. They were discovered in 1955 by Christian de Duve [6,7]. They are small vesicles with a diameter of 0.25–1 μm, surrounded by a single specific lipid-protein membrane with a thickness of 7–10 nm [8,9]. One of the main components of the lysosomal membrane are proteins belonging to glycoproteins, accounting for more than 50% of all membrane proteins—the most abundant of which are LAMP-1, LAMP-2 and LIMP-2 proteins [10]. An important protein found in the lysosomal membrane is the proton pump belonging to the H^+^-ATPase family, which ensures the acidic pH inside the lysosome [11,12]. These proteins are mostly highly glycosylated, which protects them from the action of proteases [8,9,10,11,12,13]. Also located in the lysosomal membrane are transport proteins that allow the final products of digestion to be transported to the cytosol for excretion or reuse by the cell [12,14,15].

Lysosomes contain more than 50 different hydrolytic enzymes (acid hydrolases), such as proteases, nucleases, glycosidases, lipases, phospholipases, phosphatases, peptidases and sulfatases [14,16], which are synthesized in the rough endoplasmic reticulum, then modified and transported to primary lysosomes [14,17,18]. The optimal pH value of 4.5–5.5 for lysosomal enzymes is determined by the presence of the already mentioned V-type ATPase (H^+^-ATPase), which is a complex that transports in an ATP-dependent manner protons from the cytosol to the lysosome lumen [12,19,20]. The lysosomal membrane separates the lysosomal enzymes from the cytoplasm; if the membrane is damaged and the lysosomes’ contents get into the cytoplasm (pH = 7.2–7.3), the hydrolases lose their lytic activity but practically do not damage the other cytoplasm components [14]. In terms of morphologically, lysosomes are heterogeneous organelles and are electron-dense structures containing a membrane-enveloped amorphous matrix. Depending on the type of cell and its function, lysosomes vary in size, shape and distribution in the cell [17].

The main function of lysosomes is to break down proteins, nucleic acids, lipids and sugars with the participation of lysosomal hydrolases. Lysosomes are responsible for the digestion and recycling of spent or abnormal cellular macromolecules and redundant organelles along with part of the cytoplasm, which occurs by autophagy. They also take part in the digestion of extracellular material supplied by endocytosis and phagocytosis. Hence, the lysosomes are considered to be structures responsible for regulating cellular homeostasis [16,21,22]. Nowadays, lysosomal enzymes, mainly cathepsins, are also assigned other functions, including a role in bone remodeling, prohormone processing, angiogenesis, cell death and cancer cell invasion [23].

Autophagy is a process necessary to maintain cellular homeostasis and occurs as three different types: macroautophagy, microautophagy and chaperone-dependent autophagy. Microautophagy is a non-selective process in which fragments of the cytoplasm destined for degradation are directly absorbed into the lysosome by invagination of the lysosomal membrane [24]. On the other hand, chaperone-dependent autophagy is a highly selective process in which only proteins marked with a specific signal are degraded in lysosomes [25]. Due to its selectivity, this process can also play a regulatory role in various cellular processes and in the modulation of intracellular levels of enzymes or transcription factors. Currently, it is believed that this process plays an important role in the development of numerous diseases, including neurodegenerative diseases and cancer [26].

The best-known type of autophagy is macroautophagy. In the process of macroautophagy, with the participation of specific structures called autophagosomes, both cytoplasmic macromolecules and whole cellular organelles are degraded, and the biologically active monomers formed in this process are used to maintain cellular homeostasis [24]. This process also plays a protective role in stressful conditions. Macroautophagy (also called autophagy) is a multi-stage process controlled by a number of proteins, including protein expression products of genes from the ATG family [27]. Macroautophagy was considered a non-selective process, but in recent years it has been shown that it can also be a selective process when only specific cellular components are degraded [24]. One such example is the degradation of damaged mitochondria, referred to as mitophagy. Such selective degradation of the mitochondria protects cells from the process of apoptosis and is one of the essential elements of controlling the level of ROS in the cell [28]. In the process of selective autophagy (nucleophagy), damaged parts of the cell nucleus and even whole nuclei are degraded [29]. Selective degradation can also affect abnormal lysosomes; if they are not repaired, then they are labeled with ubiquitin to initiate the process of lysophagy [30,31].

In recent years, there has been a lot of research showing that cell fate is often determined by a functional link between the processes of autophagy and apoptosis. Their paths are closely related and regulated. The main link between these two processes is a protein necessary for the initiation of autophagy, i.e., beclin 1, which interacts with proteins from the Bcl-2 family that regulate apoptosis [32,33].

Also involved in the regulation of autophagy are caspases (the main execution proteins of the apoptosis process), which, showing the ability to proteolyze beclin 1, inhibit the autophagy process with the simultaneous activation of proapoptotic signals. Proteolysis results in the formation of N- and C-terminal fragments, which, by interacting with mitochondria, induce proapoptotic signals in cells [34]. In a similar way, apoptosis is activated as a result of calpain-mediated Atg5 protein proteolysis involved in the formation of autophagosomes and whose fragments, after moving into mitochondria and interacting with Bcl-XL proteins, activate apoptosis. In the processes of autophagy and apoptosis, an important role is played by the p53 protein (suppressor of neoplastic transformation), which both inhibits and activates autophagy. It has been shown that as a result of the action of a stress factor (genotoxic stress), autophagic processes are induced by activating the transcription of the target gene p53, which is regulated by damage to the autophagy modulator (DRAM) that promotes the formation of autolysosomes. At the same time, it was shown that p53 can suppress autophagy by reducing the level of the LC3 protein that controls the degradation process [35,36].

The relationship between autophagy and lysosomal membrane permeability is also noteworthy. Karch’s team, conducting research on mouse embryonic fibroblasts, showed that Bax/Bak1-deficient cells are resistant to cell death—the mechanism of which is related to lysosomal membrane permeability—and that targeting Bax proteins to lysosomes restores autophagic cell death. The monomeric mutant form of Bax is therefore sufficient to increase the permeability of the lysosomal membrane and restore autophagic death in cells with double Bax/Bak1 elimination. It has also been shown that Bax/Bak1-deficient cells treated with a lysosomotropic detergent increased lysosomal membrane permeability and restored autophagic cell death. This suggests that Bax/Bak proteins, by directly affecting the permeability of cell organelle membranes, have the ability to interfere with the main forms of cell death [37].

## 3. Role of Lysosomes in Cancer Cells

Rapidly dividing cancer cells are highly dependent on the efficient functioning of lysosomes, and cancer cells undergoing progression are characterized by large changes in the lysosomal compartment compared to normal cells [16]. The expression of these changes includes the redistribution of lysosomes from the perinuclear space to the peripheral part (which may be related to the presence of acidic pH in the extracellular space) and the modification of the number and volume of lysosomes, as well as an increase in the expression, secretion and/or activation of lysosomal enzymes including cathepsins [16]. Most of these changes are closely correlated with invasive growth, angiogenesis and drug resistance [38].

Lysosomes may play a dual role in cancer development [16,39,40,41,42,43], because they can both exert an effect on invasive tumor growth and vascular development [16,44,45], as well as protect tumor cells from the effects of certain chemotherapeutic drugs, hence they may contribute to the development of drug resistance [16,44,45]. Lysosomes contribute to the maintenance of cancer cell proliferation, which may prevent oncogene-induced aging (OIS) [46]. In cancer cells, during aging resulting from the action of oncogenes (OIS), there is an increase in, among other things, the oxidative metabolism associated with changes in chromatin structure. This can result in the passage of chromatin fragments from the cell nucleus into the cytoplasm, where their degradation in cancer cells is associated with an increased level of autophagy and the increased synthesis of lysosomal enzymes. This contributes to the maintenance of their function and translates into a slowing of the aging process [47]. Lysosomes as organelles constituting an intracellular storehouse of calcium ions are also involved in many cellular processes, and the disturbance of calcium homeostasis is correlated with various diseases. This is due to the fact that Ca2^+^-permeable mucolipin channels with transient receptor potential (TRP) (TRPML, TRPML1-3) integrate the processes of cell growth, division and metabolism. Hence, the dysregulation of TRPML activity plays an important role in cancer development [48]. In the tumor microenvironment, activated autophagy is used by cancer cells to digest redundant or damaged proteins and cellular organelles, thereby meeting increased energy and nutrient requirements [48]. It has been shown that by releasing intracellular calcium, TRPMLs are also involved in the regulation of lysosome function and autophagic processes [49]. This is related to the fact that TRPML1 activates, among other things, the calmodulin (CaM)/CaMKKβ/AMPK pathway (promoting autophagosome formation) and CaM/CaN/TFEB (protein delivery to lysosomes), maintains mTORC1 activity (preventing tumor cell death and promoting lysosome modification), enhances lysosome’s degradation functions and Syt7-dependent lysosomal exocytosis (release of hydrolases, ATP and H^+^ into the extracellular space). This contributes to changes in the tumor microenvironment, the degradation of extracellular matrix (ECM) components, and consequently enhanced tumor progression [48]. Cancer cells, in order to increase invasiveness and metastasis, also use autophagy to degrade E-cadherin (a protein involved in the interaction between cells), which contributes to the induction of epithelial-mesenchymal transition (EMT) [50,51] and enables cells to suppress their epithelial features by changing to mesenchymal. This change makes solid tumors more malignant, thereby increasing their invasiveness and metastatic activity [52,53].

It has also been shown that in the process of transformation, cancer cells can use lysosome-centered pathways to avoid the effects of chemotherapy. The involvement of lysosomes in the process of chemoresistance is related to, among other things, the action of P-glycoprotein (Pgp), which belongs to the B subfamily of ATP-binding protein membrane transporters and which is responsible for the ejection of various types of substances from the cell, including drugs [54]. Recently, it has been reported that lysosomal overexpression of P-gp is observed in resistant cancer cells, which is due to the fact that this glycoprotein is incorporated into lysosomal membranes during recycling rather than redistributed after de novo synthesis [55,56,57]. It has been shown that cancer cells expressing MDR multidrug transporters can efficiently remove lysosomotropic ionizing drugs that diffuse into the cytosol or are sequestered in lysosomes, examples of which include doxorubicin, daunorubicin, vinblastine, vincristine, or cisplatin [58]. It has been found that inducing resistance to the above-mentioned cytostatics occurs when they are Pgp substrates and must ionize at lysosomal pH, resulting in their sequestration, entrapment in lysosomes, and then their release from the cell by exocytosis [55]. Their accumulation in lysosomes is mainly due to ion traps or active transport to the lysosomes [59].

Since cancer cells, compared to normal cells, are characterized by weaker lysosomal membranes, they can be selectively sensitized to various types of cell death, of which apoptosis and autophagy are of great therapeutic importance [24]. Numerous data indicate that autophagy plays a dual role in the process of carcinogenesis [60].

Autophagy can inhibit the process of carcinogenesis by limiting the survival of cancer cells and initiating cell death. However, it can also promote the development of transformed cells, as indicated by numerous studies that prove that in cancer cells there is a reprogramming of the autophagy process associated with the activation of oncogenes and the inactivation of tumor suppressor genes [61]. Autophagy may therefore be a mechanism that benefits cancer cells and tumor development also by promoting increased nutrient delivery to cancer cells, resulting in increased survival [61].

The increased permeability of lysosomal membranes is another feature of lysosomes in cancer cells that is very important from the point of view of anticancer therapy [16,23,42,44,62,63]. According to numerous works [16,42,44], the increased permeability of lysosomal membranes may be important in improving the transport of anticancer drugs. Also, the increased metabolism of cancer cells, and thus the accelerated turnover of proteins including iron-containing proteins, can lead to the accumulation in lysosomes of iron (II) ions, which, reacting with hydrogen peroxide, increase the production of hydroxyl radicals (Fenton reaction), which can induce lysosomal cell death [39,64].

Therefore, the aim of numerous studies in recent years has been lysosomes, the types of death associated with them, understanding the mechanisms of the chemoresistance of cancer cells and the possibilities of overcoming it.

## 4. Lysosomal Cell Death—A New Strategy for Anticancer Therapy

For many years, it was thought that caspases played a major role in the process of apoptosis, and the role of lysosomes in apoptosis was supposed to be limited only to digesting the contents of apoptotic bodies. In 1996, Deiss’ team showed that cathepsins, degradative enzymes that are now considered important mediators of apoptosis, are also involved in the so-called “lysosomal pathway of apoptosis”. According to the literature, the destabilization of lysosomes is a critical moment not only for these organelles but also for cell function, as a moderate release of enzymes from lysosomes leads to apoptosis, while the rupture of lysosomal membranes can lead to necrosis [22,65,66,67] and to neurodegenerative diseases such as Alzheimer’s, Parkinson’s or myocardial infarction [62,63,65,68,69,70,71,72,73,74].

Cathepsins are among the most abundant lysosomal proteases identified in acidic endo/lysosomal compartments, where they play an important role in intracellular protein degradation, energy metabolism and immune responses [75]. Cathepsins have been divided into three subgroups depending on the presence of active amino acids: cysteine (B, C, F, H, K, L, O, S, V, X and W), aspartate (D and E) and serine (A and G). The discovery that cathepsins are secreted outside the lysosome and remain active in this state has resulted in numerous studies revealing the many versatile functions of cathepsins in both extra-lysosomal locations, including the cytosol and the extracellular space [75]. Cathepsins show the highest activity at the slightly acidic pH of the lysosome. However, some of them have also been found to be active beyond their optimal range (pH = 5) [76]. Cathepsins are enzymes that play an important role in the autophagy process, of which cathepsins D and L are responsible for the degradation of autolysosome content and the regulation of lysosome and autophagosome populations. An important role in the autophagy process is also played by cathepsin B, which, by degrading the MCOLN1/TRPML1 calcium channel in lysosomes, leads to the suppression of the TFEB transcription factor, which results in inhibition of the expression of proteins associated with autophagy [77]. It was also shown that cathepsin S plays a major role in the fusion of autophagosomes and lysosomes, the deficiency of which leads to the accumulation of autophagosomes [78].

In addition to degradation processes, cathepsins are also involved in another type of cell death—lysosomal death. This process is initiated by increasing the permeability of the lysosomal membrane (LMP), which leads to the release of cathepsins into the cytosol and the start of the cell destruction process [79]. LMP is induced by various factors, which are described in the next section.

In the process of cell death, a significant role has been attributed to cathepsin D, L and B [76,80,81]. Currently, the important role of cathepsins in cancer processes is more frequently indicated, which is related to their localization in different cellular compartments. Numerous papers have described the overexpression of cathepsins in breast, colorectal, gastric, lung, prostate, thyroid and brain cancers. Therefore, cathepsins are often attributed prognostic significance [82]. Cathepsins released by exocytosis into the extracellular space promote tumor progression, metastasis and angiogenesis [41,82]. Cathepsin B stimulates tumor angiogenesis by degrading the extracellular space and inactivating the tissue inhibitors of the extracellular matrix (TIMP-1 and -2) [41]. Cathepsin D secreted in a catalytically inactive form affects progression and angiogenesis [83,84], whereas the response to the action of cathepsin S is the stimulation of tumor angiogenesis [85].

### Mechanism of Lysosomal Cell Death

The pro-apoptotic effect of cathepsins is the result of the increased permeability of the lysosomal membrane, which leads to the release of lysosomal proteases into the cytosol and, consequently, to the induction of apoptotic death; this process occurs in various ways (Figure 1) [70]. The increased permeability of the lysosomal membrane is therefore an important step in the lysosomal cell death pathway, comparable to the damage to the mitochondrial membrane in the process of apoptosis [81]. Acidic pH is a factor in the activity of cathepsins, and their activity has been found in the cytosol, where the ability of the enzymes to process cytosolic substrates has been demonstrated. This mechanism can probably be explained by the acidification of the cytosol by hydrogen ions that were released from the inside of the lysosomes during membrane damage [63,86]. Cathepsins released from lysosomes, as initiators of lysosomal cell death, affect the activation of caspase 8 and Bid protein, which in turn contribute to the activation of Bax/Bak protein. Cathepsin B also contributes to the activation of the pro-apoptotic Bcl-2 family protein, PLA2 protein, caspase 2, as well as the activation of sphingosine kinase 1 and ceramide [23,70], which leads to cell death. Unlike caspases, which require activation to initiate the process of apoptosis, cathepsins are active in the cytosol right after being released from the lysosomes [22]. The confirmation of the effect of cathepsins on the activation of the apoptosis process is the microinjection of cathepsin D into the cytosol of the cell, resulting in the induction of cytochrome c [87].

As a consequence of the increased permeability of the lysosomal membrane in cancer cells, the death pathway may occur, which runs independently of executive caspases, during which cathepsins leak and cytosol acidification and caspase-2 activation occur [88].

However, it has been shown that cancer cells can inhibit both the apoptotic and lysosomal pathways in order to differentiate and proliferate [23]. An increased activity of phosphatidylinositol 3-kinase (PI3K) in cancer cells may affect many features of lysosomes, such as maturation rate, size, activity and stability [43,44]. On the other hand, the inhibition of the activity of phosphatidylinositol 3-kinase contributes to the change of the cell death pathway dependent on caspases to dependent on cathepsins [43]. Cancer cells can also protect themselves from lysosomal membrane damage by translocating a heat shock protein (Hsp 70) to their membrane. This protein contributes to the accumulation of enlarged lysosomes with stable membranes in the cells, which is done by increasing the activity of acid sphingomyelinase. The use of drugs that inhibit kinase and block the localization of Hsp 70 protein in the membrane of lysosomes can sensitize cancer cells to anticancer drugs [23,44,89].

Under conditions of cellular homeostasis, the degradation of molecules, cell organelles and cell fragments occur by macroautophagy involving the fusion of primary lysosomes containing a set of lysosomal enzymes (hydrolases) with autophagic vacuoles (autophagosomes). In contrast, sequestration of molecules or cell fragments into lysosomes occurs by microautophagy.

LMP inducers cause a slight leakage of cysteine proteases (cathepsins) into the cytoplasm, which activate pathways leading to cell death. Complete disruption of the lysosomal membranes leads to a massive release of lysosome components into the cytoplasm, causing necrosis. LMP-inducing agents include lysosomotropic compounds, some antihistamines, immunosuppressants, DNA-damaging drugs, chemotherapeutics, photosensitizers, plant compounds (anthraquinones, vinca alkaloids, taxoids). List of abbreviations: LMP—lysosomal membrane permeability, ROS—reactive oxygen species, MOMP—mitochondria outer membrane permeabilization.

## 5. Lysosomal Membrane-Damaging Agents and Their Use in Anticancer Therapy

The available literature describes a number of agents that cause modulation of the lysosomal compartment, an increase in lysosomal enzyme activity (including cathepsins) and an increase in the lysosomal membrane permeability (LMP) including its damage, which can lead to the release of lysosomal proteases and their proapoptotic effect (Figure 2; Table 1).

### 5.1. Lysosomotropic Compounds

Lysosomotropic compounds have a high affinity for accumulating in lysosomes (lysosomotropism), and many of them have therapeutic activity (cationic amphiphilic drugs—CAD) [115]. The CAD family covers a wide spectrum of compounds, including dozens of approved and already-used drugs to treat a wide range of diseases. These compounds are used as bactericides, fungicides, antimalarials (chloroquine), antipsychotics (chlorpromazine, thioridazine and aripiprazole), antihistamines (astemizole, clemastine, desloratadine, ebastine, loratadine and terfenadine) or antidepressants (desipramine, imipramine and clomipramine) [86,116,117].

Despite belonging to different pharmacological classes, the aforementioned compounds share the same physicochemical properties. Common to all CADs is their amphiphilic nature, which is due to their hydrophobic ring structure and hydrophilic side chain containing cationic amino groups [118]. According to de Duve’s (1974) definition [119]), lysosomotropic molecules are weak organic bases that have the ability to penetrate the phospholipid bilayer and accumulate in low pH cellular compartments, i.e., lysosomes or vacuoles [120,121]. These compounds are protonated and trapped inside the lysosomes, and their diffusion back into the cytosol is significantly impaired. During accumulation, these drugs strongly stress both lysosomal and other biological membranes inside the cell, due to the close binding of the molecule to the surface of the phospholipid bilayer [86,121,122]. The effect of lysosomotropic drug accumulation on cells is characterized by the promotion of certain changes such as cytoplasmic vacuolization, an increase in the number and size of lysosomes, inhibition of their enzymes and accumulation of undecomposed material, leading mainly to phospholipidosis [115,123]. The high ability to accumulate weak bases in highly acidic compartments is due to the difference in pH between the interior of the lysosome (pH 4–5) and late endosome (pH 5–6) and the cytosol (pH 6.8–7.4) [121]. For potent lysosomotropic drugs, it has been estimated that 50–70% of the compound accumulated intracellularly is stored in lysosomes and endosomes, leading to extreme concentrations in these compartments [115]. This especially refers to the lysosomes of cancer cells, where the pH is lower than in normal cells [124]. The increased sensitivity of cancer cells to certain lysosomotropic drugs has become a promising approach to selectively destroying cancer cells [116]. Due to their unique ability to penetrate and accumulate in lysosomes, lysosomotropic compounds may contribute to the enhanced efficacy of cytostatics used in treatment. Therefore, attempts are now being made to include them into anticancer therapy in order to eliminate the significant problem of multidrug resistance [107].

### 5.2. Lysosomotropic Detergents

An interesting group of substances are lysosomotropic detergents (LDs), which combine features of lysosomotropism with detergent activity [125]. These are lysosomotropic amines that are capable of passive diffusion across cell membranes and accumulation in lysosomes until their concentration is sufficient to solubilize the lysosomal membrane. Unlike other detergents that kill cells by acting on the cell membrane, lysosomotropic detergents act mainly from inside lysosomes [86]. The lysosomotropic properties of some detergents represent a promising strategy for anticancer treatment that involves promoting apoptosis in cancer cells by releasing cathepsins, which thereby activates the lysosomal cell death pathway [22]. In addition, LDs may find application in the drug delivery system because, being amphiphilic molecules, they spontaneously organize themselves into aggregates that can serve as carrier systems [86].

An example of endogenous lysosomotropic agents is sphingosine, generated in the lysosomal membrane by the sequential action of acid sphingomyelinase (aSMase) and acid ceramidase, which convert sphingomyelin to ceramide and then to sphingosine [126]. Sphingosine can act as a typical lysosomotropic detergent, and the cell death induced by its action involving LMP and the leakage of hydrolytic enzymes into the cytosol occurs by two different mechanisms depending on the concentration of the compound. In studies on Jurkat T-lymphoma cells and J774 cells after administration of sphingosine at low to moderate concentrations, the partial rupture of lysosomes was shown to precede caspase activation, as well as a change in mitochondrial membrane potential. High concentrations of sphingosine, on the other hand, caused the extensive rupture of the lysosomes and resulting necrosis without preceding apoptosis or caspase activation [127]. Sphingosine has also been shown to mediate the TNF-α-induced permeabilization of the lysosomal membrane and the resulting programmed cell death. Studies on a variety of hepatoma cell lines (McNtcp.24, SK-HEP-1, Hep G2, and Huh7) confirmed a marked increase of sphingosine in induced apoptosis. Moreover, the accumulation of sphingosine and ceramide in lysosomes modulated cathepsin B and D activity and induced LMP [128,129,130].

Damage to the lysosomal membrane may also result from the inhibition of acid sphingomyelinase (ASM), which, as demonstrated by Petersen’s team in 2013 [131], moves in an ATP-dependent manner protons from the cytosol to the lysosome lumen, which may be an important mechanism of action of siramesine. Siramesine is a sigma-2 receptor antagonist that was originally developed for the treatment of depression and anxiety, but with demonstrated low efficacy. Studies have shown, however, that siramezin has a stronger effect on cancer cells [107]. The receptors for this ligand are overexpressed in a number of cancers: breast, pancreas, nervous system, bladder and lung, and this may provide the basis for its use in radiation therapy for cancer imaging [88]. According to available data, siramesine is the most potent anticancer agent among σ-ligands. It has the property of triggering a cancer cell-specific alternative cell death pathway. Acting as a lysosomotropic detergent, siramesine induces cancer cell death in MCF-7 breast cancer cells because it rapidly localizes to lysosomes and neutralizes the pH of lysosomes, causing direct disruption of the lysosomal membrane and leakage of cathepsins into the cytosol. Death induced by siramesine is independent of the main modulators of classical apoptosis, i.e., p53, Bcl-2 and caspases [107]. In addition to lysosome damage, siramesine induces oxidative stress in cells while showing cytoprotection through the massive accumulation of autophagosomes. The accumulation of non-degradable material in damaged lysosomes and progressive oxidative stress can further increase lysosome rupture and lead to cell death [107]. Therefore, for anticancer therapy, a combination of siramesine and inhibitors of autophagosome formation has been suggested [107,131]. In a study by the Groth–Pedersen team in 2007 [90], it was confirmed that the administration of siramesine to tumor cells simultaneously encumbered with vincristine increased the cytotoxic effect of this alkaloid.

The strategy of targeting lysosomes by siramesine described in Dielschneider’s work may also be a new form of treatment for chronic lymphocytic leukemia (CLL cells). Literature data [108] show that siramesine induced in CLL cells (characterized by high toxicity and drug resistance) permeability of lysosomal membranes, lipid peroxidation, loss of mitochondrial membrane potential and release of reactive oxygen species. At the same time, it was shown that the CLL cells had numerous lysosomes, an increased expression of sphingosine-1-phosphate phosphatase (SPP1), and high levels of sphingosine-induced permeabilization of the lysosomal membrane and lysosomal cell death [108].

### 5.3. Immunosuppressant Drugs

Another group of compounds that affect the lysosomal compartment are some immunosuppressant drugs. One of these is FTY720, a sphingosine-1-phosphate (S1P) analog, which is used to treat multiple sclerosis. FTY720 has also shown anti-tumor activity, but the molecular mechanisms are not fully clear. A study by Min and Kwon in 2020 on glioma cells (U251MG, U87MG and U118MG) showed FTY720 accumulating in the lysosome-induced permeabilization of the lysosomal membrane. On the other hand, the inhibition of LMP by HSP70 and the cathepsin inhibitors blocked FTY720-induced cell death, suggesting that this compound induces glioblastoma cell death after permeabilization of the lysosomal membrane [104].

### 5.4. Antihistamine Drugs

Research on the assessment of the potential anticancer mechanisms of antihistamines is part of the current canon in the fight against cancer, which consists in the recession of already approved drugs with proven safety and which translates into shortening the time to obtain a drug for oncological treatment. Lysosomotropism is a feature of some representatives of antihistamines, characterized by different physicochemical properties related to their cationic-amphiphilic nature (CAD antihistamines) [105].

According to the study, astemizole, loratadine and ebastine have a high volume of distribution, which translates into the efficient distribution of the drugs into tissues, and because they are weak bases, they accumulate more effectively in acidic tumors than in healthy tissues with a neutral pH. The study authors also suggest that the use of the aforementioned CAD antihistamines in combination with chemotherapeutics can sensitize cells to chemotherapy and reverse multidrug resistance, thereby effectively enhancing the anti-tumor response. Therefore, it is important to expand studies to evaluate other representatives of the antihistamine group of drugs. In our study [106], we showed that azelastine hydrochloride (a second-generation LPH drug) at concentrations of 15–25 µM induces degradative processes and significantly intensifies vacuolization changes, as well as increases the activity of cathepsin D and L and LC3-II protein. The consequence of azelastine at concentrations of 45–90 µM was a marked promotion of apoptosis through the activation of caspase 3/7 and the inactivation of Bcl-2 protein with a concomitant increase in ROS levels. In contrast, a study by Kim’s team in 2019 showed that azelastine, like desloratadine, has the property of sensitizing KBV20C cells (a highly antimitotic-resistant line) to vincristine in combination administration [132].

### 5.5. Thiosemicarbazone Analogs

The research of Richardson’s team shows that the lysosomal pathway can also be selectively activated by metal chelating compounds from the thiosemicarbazone di-2-pyridylketones (DpT) group, which are characterized by a strong and selective anticancer effect [113]. The aforementioned team of researchers also showed that DpT analogs, i.e., di-2-pyridyl ketone 4,4-dimethyl-3-thiosemicarbazone (Dp44mT) and di-2-pyridyl ketone 4-cyclohexyl-4-methyl-3-thiosemicarbazone (DpC), generate stress oxidative activity in cancer cells, induce permeabilization of the lysosomal membrane and increase cytotoxicity, which may contribute to overcoming resistance to anticancer drugs [133]. According to previous studies [113], the targeting of Dp44mT on lysosome integrity was conditioned by copper binding and was essential for Dp44m’s potent anticancer activity, while co-incubation with non-toxic copper chelators significantly attenuated the cytotoxicity of this compound. It has been shown that thiosemicarbazones, in addition to affecting lysosomes, have the ability to activate other types of cell death, such as apoptosis and autophagy, and also inhibit c-Met oncogene expression through mechanisms that include lysosomal degradation [134].

### 5.6. Photosensitizers

Compounds that are used in cancer phototherapy also play an important role in lysosomal cell death; an example is N-aspartylchlorin (e-6, NPe6), a photosensitizer with a very high affinity for cancer cells and which accumulates in lysosomes and contributes to the generation of singlet oxygen. The reactive oxygen species generated by the photodynamic process cause rapid damage to lysosomes, leading to the rapid release of cathepsins into the cytosol, the activation of the Bax protein that initiates the mitochondrial apoptosis pathway, and procaspases 3 and 9 [102,135]. A study by Chiarante’s team [103] showed that the lipophilic photosensitizer phthalocyanine Pc9 encapsulated in T1107 polymeric micelles promoted the induction of caspase-dependent apoptotic cell death. This compound, after accumulating mainly in lysosomes and the endoplasmic reticulum, promoted the induction of reticulum stress and the permeabilization of lysosomal membranes, resulting in the release of cathepsin D into the cytosol and a 50% decrease in Hsp70, which is considered a lysosomal stabilizer.

### 5.7. Plant Compounds

Compounds that induce LMP include plant antimitotic drugs, i.e., vinblastine, vincristine, vinorelbine (*Catharanthus roseus* alkaloids) [90,136], paclitaxel and docetaxel from *Taxus brevifolia* [62,91,137], cisplatin, etoposide, camptothecin (DNA-damaging drugs) [62], and some antibiotics, including norfloxacin, ciprofloxacin, nanomycin [62,138], gentamicin [100] and azithromycin [109].

Factors damaging the lysosomal membrane are also other compounds of plant origin, such as resveratrol [92] and its analog pterostilbene [139], ginsenoside Rh2 from ginseng [93], and triptolide from the Chinese herb Tripterygium Wilfordii Hook F [94].

Our studies also show that compounds modulating the activity of the lysosomal system are anthraquinones, compounds isolated from aloes of various species (*Aloe barbadensis* Mill., *Aloe arborescens* Mill.), as well as from *Frangula alnus* Mill., or from rhubarb (*Rheum palmatum* L., *Rheum officinale* Baill.), which is used in traditional Chinese medicine [140,141,142,143]. Anthraquinones are compounds with multidirectional biological activity, exerting antibacterial [144], antifungal, antiviral (including inhibiting the replication of the SARS-CoV-2 coronavirus) [145], cleansing, anti-inflammatory, immunoregulatory and antihyperlipidemic effects [146]. They also exhibit anticancer activity, mainly through DNA damage, cell cycle arrest or pro-apoptotic activity. The anticancer mechanism of anthraquinones is also based on the inhibition of tumors through paraptosis, autophagy, radiosensitization and overcoming chemoresistance [147]. One of the mechanisms is also the generation of reactive oxygen species, which are an important group of endogenous inducers of lysosomal membrane permeability [148]. They are formed not only as a result of certain drugs, but also heavy metals or ionizing radiation [100,149]. As a consequence of these various agents, damaged mitochondria can generate increased amounts of reactive oxygen species, and their effects on lysosomal membranes can result in the release of lysosomal contents into the cytoplasm. It has also been shown that reactive oxygen species can activate phospholipase A2 (PLA2), which can cause destabilization and increased permeability of the lysosomal membrane [23,41,67,70,72,79,82,87,138,149,150,151,152,153,154]. The free radicals show the ability to interact with free intralysosomal iron, creating highly reactive hydroxyl radicals in the Fenton-type reaction, which contributes to the induction of LMP through the ongoing lipid peroxidation of lysosomal membranes, resulting in the formation of lipofuscin and further damage to the lysosomal membrane proteins [79,155].

On the other hand, Zhao in 2003 showed that induction of lysosomal membrane permeability may also precede mitochondrial damage, as lysosomal enzymes may act on the mitochondrial membrane and promote the generation of reactive oxygen species, which in turn may further sensitize the lysosomal membrane to damage [156].

Examples of anthraquinones that exhibit cytotoxic effects against the lysosomes of tumor cells (HeLa line) are aloe-emodin [95] and emodin [96]. The consequence of ROS generation in cells by the tested compounds was the induction of LMP, and cathepsins D and L released from lysosomes contributed to the activation of executive caspases 3/7, which in turn translated into a decrease in Bcl-2 protein expression and an intensification of apoptotic processes. The anthraquinone that induces changes in the lysosomal compartment is also rhein, which is the main metabolite of diacerein, a drug used in osteoarthritis. We have documented rhein-induced degradation processes in HeLa cells expressed by an increase in the number of primary lysosomes, autophagosomes and autolysosomes. High rhein concentration generated ROS, which induced LMP as expressed by fluorescence quenching of the non-enzymatic marker of lysosomes, acridine orange, which correlated with a reduced neutral red uptake by lysosomes and increased cathepsin D and L activity in the extra-lysosomal fraction. The mechanism of rhein’s pro-apoptotic effect was associated with an increase in caspase 3/7 activity and a decrease in Bcl-2 protein expression. In parallel, we showed that the pre-incubation of cells with chloroquine inhibited rhein-induced autophagy and contributed to increased cytotoxicity against HeLa cells. Our results confirmed that rhein, by inducing changes in the lysosomal system, indirectly influences apoptosis of HeLa cells, and its combined action with autophagy inhibitors may be an effective form of anticancer therapy [97].

### 5.8. Combination Therapy

Currently, in anticancer therapy, attention is paid to combining various types of drugs with anticancer drugs in order to achieve better treatment results. This approach has the potential to reduce drug resistance while providing anti-cancer therapeutic benefits that include the reduction of tumor growth and metastatic potential, the arrest of mitotically active cells, the reduction of cancer stem cell populations, and the induction of apoptosis [157]. Such an example is the enhancement by azithromycin (a macrolide antibiotic, belonging to the group of so-called azalides, with bacteriostatic activity) of the cytotoxic effect of doxorubicin, etoposide and carboplatin [109]. Treatment of cells with doxorubicin increased lysosomal biogenesis through the activation of TFEB and led to lysosomal membrane damage (as confirmed by galactin-3 puncta assay) and a leakage of lysosomal enzymes into the cytoplasm. In contrast, azithromycin treatment blocked autophagy, as expressed by the accumulation of lysosomes in cells. It was shown that the combined effect of doxorubicin and azithromycin integrated with the growth of damaged lysosomes led to a pronounced permeabilization of the lysosomal membrane, inducing apoptosis. Based on the results, the authors believe that the simultaneous action of DNA-damaging drugs and azithromycin is a promising strategy for the treatment of non-small-cell lung cancer (NSCLC) through the induction of LMP [109].

Lysosomal membrane permeabilization has also been used to restore the sensitivity of lung cancer cells (A549cisR) with acquired chemo-resistance to cisplatin, and is commonly used in advanced lung cancer therapy [98]. According to the study, exposure of A549cisR cells to cisplatin did not induce an apoptotic effect. In contrast, incubation of A549cisR cells with chloroquine promoted LMP in the cells, as was evident by immunofluorescence microscopy through leakage of FITC-dextran into the cytosol, as well as confirmed by immunoblotting technique through increased cytosolic cathepsin D signal [158]. The authors of the study suggest that synergism between the two drugs may be important in planning new therapies.

A combination of microtubule-disrupting drugs such as vincristine with lysosome-destabilizing compounds, an example of which is the described siramesine, may be of great importance for cancer therapies. The study showed that combining non-toxic concentrations of vincristine with siramesine effectively induced massive death of cells resistant to the alkaloid (MCF-7 line) [90].

Combination therapy based on a lysosomal system has attracted attention in the context of the use of nintedanib (BIBF1120), which, as a small molecule tyrosine kinase inhibitor (TKI), inhibits the activity of receptor tyrosine kinases (TKs) that include vascular endothelial growth factor receptor (VEGFR 1–3), platelet-derived growth factor receptor (PDGFR-α i-β) and fibroblast growth factor receptor (FGFR 1–3) [159]. Because nintedanib effectively interferes with the proliferation and migration of lung fibroblasts, it is used in the treatment of idiopathic pulmonary fibrosis (IPF), chronic lung disease [160] and in combination with docetaxel for the treatment in non-small-cell lung cancer (NSCLC) [161].

Studies have shown that resistance to nintedanib treatment results from the fact that in lung cancer cells, this compound can be sequestered into lysosomes, where it becomes protonated and trapped. At the same time, it has been shown that the combination of the drug with bafilomycin A1 or chloroquine contributes to the reduction of its accumulation in lysosomes, which should be associated with the inhibition of acidic pH in lysosomes [111]. Nintedanib also induces autophagic activity in cells in a beclin 1-dependent and ATG7-independent manner [112].

In combination therapy with siramesine, another tyrosine kinase inhibitor, lapatinib, also shows a positive effect in the treatment of prostate cancer. In this case, the intensification of cell death was the result of increased lysosomal membrane permeability, the enhanced production of ROS, and decreased mitochondrial membrane potential. The combined effect of lapatinib and siramesine contributed to the induction of apoptotic death, PARP cleavage and reduced expression of the anti-apoptotic protein Mcl-1 [110].

## 6. Summary

One of the reasons for the failure of anticancer therapies is the drug resistance of cancer cells, which are characterized by their ability to inhibit programmed cell death by disrupting the apoptosis pathway. Numerous works show that for new therapies in oncological treatment, it is important to activate in cancer cells other apoptosis pathways leading to their death. Such a pathway may be lysosomal cell death. Currently, it is believed that the lysosomal system may be of great importance in chemotherapy because lysosomes in cancer cells are characterized by specific properties that can be used to target anticancer drugs inside them. Their high accumulation inside lysosomes can contribute to the strong sensitization of membranes, resulting in increased permeability, increased leakage of cathepsins into the cytoplasm, induction of apoptotic processes and, consequently, increased cytotoxic effects. From the point of view of modern cancer therapy, it is also extremely important to pay attention to combination therapies in which various types of drugs (including non-cancer drugs), as a consequence of their use with chemotherapeutics, increase cytotoxicity which translates into treatment effectiveness.

Understanding the exact mechanisms that induce lysosomal membrane permeability, as well as the mechanisms of action of cathepsins at the cellular level as initiators of cell death, seems to be a highly relevant task for modern oncological treatments.

## Figures and Tables

**Figure 1 ijms-24-02176-f001:**
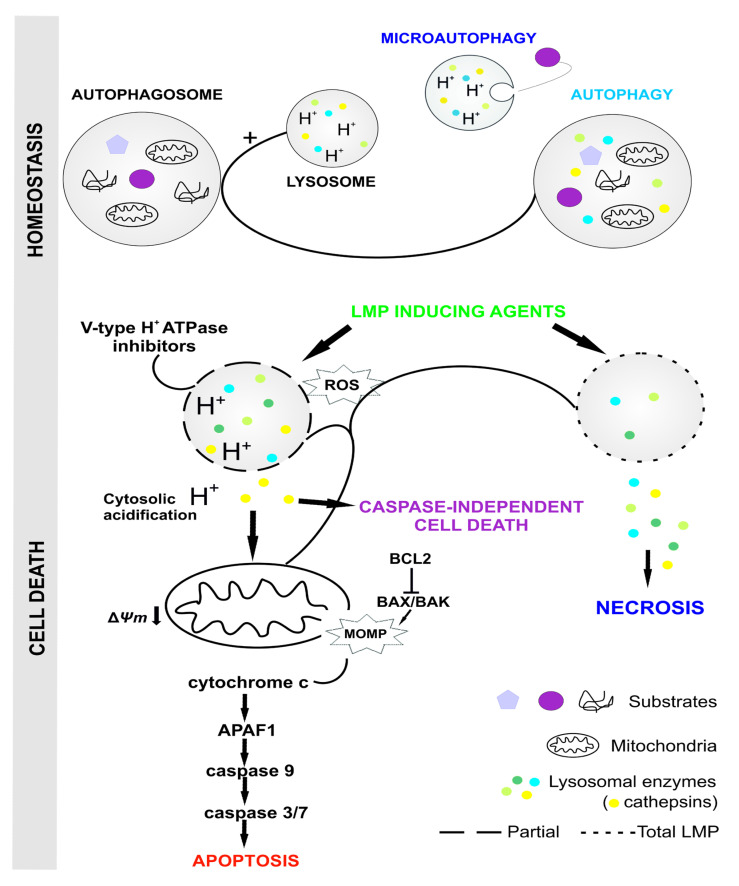
Intracellular mechanism of lysosomal membrane permeability inducing lysosomal cell death.

**Figure 2 ijms-24-02176-f002:**
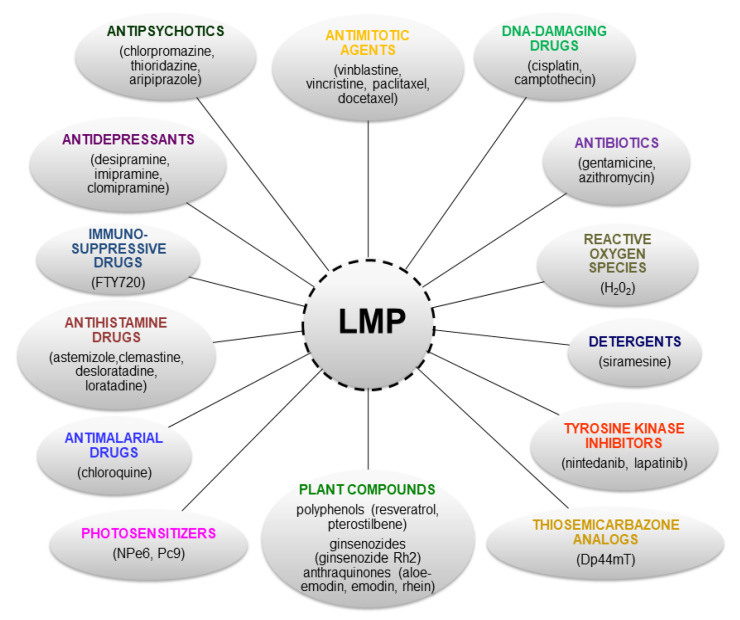
Examples of compounds inducing lysosomal membrane permeability (LMP).

**Table 1 ijms-24-02176-t001:** Summarization of compounds with anticancer activity inducing changes in the lysosomal compartment in cancer cells.

Compounds with Anticancer Activity	Cells	Mechanism of Action
Vincristine (from *Catharanthus roseus)*	HeLa	Increase the lysosomal volume and lysosomal leakage; activation of apoptosis [90]
Paclitaxel (from *Taxus brevifolia)*	NSCLC	Disruption of lysosomes; release and activation of cathepsin B [91]
Resveratrol, Pterostilbene (from grapes)	DLD1, HT29	Relocation of cathepsin from lysosomes to the cytoplasm; increase of the permeability of mitochondrial membranes, activation of caspase 3 [92]
Ginsenoside Rh2 (from Ginseng)	HepG2	LMP with the release of cathepsins to the cytosol; activation of the lysosomal-mitochondrial apoptotic pathway [93]
Triptolide (from the Chinese herb Tripterygium Wilfordii Hook F)	MCF-7	Leakage of lysosomal enzymes into the cytosol, increase permeability of the lysosomal membrane [94]
Aloe-emodin, emodin, rhein (Anthraquinones from Aloe sp. and Rhubarb)	HeLa	Increased number of lysosomes and autolysosomes; ROS generation; induction of LMP, cathepsin D and L released from lysosomes; activation of caspases 3/7, decrease in Bcl-2 protein expression, increase in apoptosis [95,96,97]
DNA-damaging drugs cisplatin camptothecin	A549cisR U-937	Autophagy activation [98] Lysosomal labilization, cathepsin B activation; increase the permeability of mitochondrial membranes and caspase activation [99]
Antibiotics gentamicin, azithromycin	LLC-PK1 U2OS, H4	ROS production, LMP [100] the loss of lysosomal acidity/function [101]
N-aspartylchlorin (e-6, NPe6)	1c1c7	Release of cathepsin D from late endosomes/lysosomes and activation of procaspase-3 [102]
Phthalocyanine Pc9 encapsulated in T1107 polymeric micelles	CT26	LMP induction, cathepsin D mediates Bid cleavage and caspase 8 activation; induces ER stress [103]
FTY720 (a sphingosine-1-phosphate (S1P) analog)	U251MG, U87MG, U118MG	Induction of vacuolization of the cytoplasm and LMP [104]
Antihistamine drugs astemizole, loratidine, ebastine, azelastine hydrochloride	KG-1 HL-60 THP-1 HeLa	Disturbance of lysosomal and mitochondrial homeostasis [105] Induction of degradation processes, increased activity of cathepsin D and L, LC3-II protein; ROS increase, activation of caspase 3/7, inactivation of Bcl-2 protein [106]
Siramesine (sigma-2 receptor antagonist)	MCF-7 CLL	Massive accumulation of autophagosomes, induction of oxidative stress, disruption of the lysosomal membrane, leakage of cathepsins into the cytosol [107] LMP induction, lipid peroxidation, loss of mitochondrial membrane potential, high ROS level [108]
Doxorubicin and azithromycin	NSCLC	Growth of damaged lysosomes, induction of LMP and apoptosis [109]
Cisplatin and chloroquine	A549cisR	Promoting LMP and cathepsin D activation [98]
Vincristine, vinblastine, taxol and siramesine	MCF-7	Microtubule disruption and destabilization of lysosomes [90]
Lapatinib and siramesine	DU145 PC3 LNCaP	LMP increase, high ROS level, loss of mitochondrial membrane potential, induction of apoptosis [110]
Nintedanib and chloroquine/bafilomycin A1	NCI-H1703 NCI-H520 DMS114	Alkalization of lysosomes (resistance inhibition), autophagy activation [111,112]
di-2-pyridylketone 4,4-dimethyl-3-thiosemicarbazone (Dp44mT)	KB31 KBV1 SK-N-MC MCF-7 DMS-53	Pgp-dependent LMP based on Cu chelation, ROS generation, increased relative lipophilicity [113,114]

## Data Availability

The data that support the findings of this study are available from the corresponding author upon reasonable request.

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
