# Peer review of "Lysosomes as a Target of Anticancer Therapy"

_ijms, 2023, doi:10.3390/ijms24032176_

Round 1
Reviewer 1 Report
Wojciech et al present a review of the literature about the pharmalogical treatment options available for cancer, regarding drug resistant. The authors addressed the lysosomal pathway as a tool to overcome this issue. Several topics are adressed, like pathophysiology of the compounds in treatment, which are of great interest to the reader, that works in this field. It is very important to have papers like this, where relies the main ideas of the topic adressed. However, I think the authors should review the paper, in order that the main ideas can reach the reader more easily. Independent from the editorial evaluation of this issue, the authors may want to consider the following suggestions for improvement:
- the english should be improved. Sometimes the sentences are confusing. For instance, Line 38, line 138-144
- add more references to suppor the ideas. For instance, Line 51
- re-check the acronyms in the texto. For instance, line 178, NR
Author Response
Responses to Review 1
- The English language has been corrected in the manuscript.
- Abstract quality improved.
- The article has been reviewed and the following changes have been made:
3a. Chapter 5 was supplemented with additional compounds inducing changes in the lysosomal compartment. Figure 2 and table 1 has been supplemented with new LMP-inducing compounds.
3b. Chapter 2 has been supplemented with a description of the process of autophagy, its types, as well as their importance in the development of cancer and the emergence of resistance to treatment.
3c. As suggested by the reviewer, the title of the manuscript was modified, the current title: "Lysosomes as a target of anticancer therapy".
3d. Modifications have been made to individual chapters and subsections of the manuscript.
3e. The article was enriched with additional literature items.
Thank you very much for the review received and for your valuable comments that contributed to improving the quality of our manuscript.
Reviewer 2 Report
In the manuscript “Lysosomal path of cell death in cancer therapy”, Trybus et al aims to summarize the current knowledge on this new form of cell death and its inducers. Overall, the manuscript is well written with a clear storyline. The English should be improved in some sentences. With regard to the topic, this review is not the first dealing with lysosomes as target for cancer therapy (e.g. PMID: 28757908, PMID: 28025106, PMID: 30071644). Still, the paper is very interesting and the abstract presents well the goal of the overall work. However, there are several issues which need to be addressed before publication of this review:
The article opens up with a quite short introduction. The introduction does not clearly explain in my opinion what is the connection between cancer resistance (main topic of the first introduction paragraph), and the final part of the section (starting around line 40) regarding lysosomal cell death. The authors should clarify better whether the study of lysosomal cell death is “important” because the induction of this cell death could overcome other cancer resistance mechanisms (such as apoptosis-resistance, mentioned earlier) and why is preferrable over paraptosis or necrosis induction, for example. In general, the authors should emphasize more the importance of this study, naming some more valid reasons. In addition, they should clarify the added value of their article in comparison to others.
The next paragraph, regarding structure and function of lysosomes, is also quite short, probably because in the last section (from line 71) is just listing the main functions of the organelle, without explaining in more details the autophagic process or the difference between the various autophagic pathways (macro and microautophagy and chaperone-mediated autophagy) which could be important for the understanding of the lysosomotropic agents effects on the organelle. In line 61, the authors state that the optimal lysosomal pH is 5; it is important to remember though that the lysosomal pH swings between 4.5 and 5.5.
In line 97 the authors state that “increasing the lysosomal permeability may (..) improve the transport of anticancer drugs”. However, it is not clear what “transport” means in the sentence; are the authors suggesting that increasing the lysosomal permeability, the lysosomotropic agents would be no longer trapped in the organelle and therefore would have more access to their cytoplasmic targets?
In paragraph 3 the authors talk about cathepsins, lysosomal enzymes able to work just in acidic pH conditions. Further on in the paper, the authors claim that these enzymes are the main drivers of lysosomal cell death (line 374), because they induce the activation of cell-death promoting proteins. Is this induction simply due to the cathepsins dislocation in the cytoplasm, perceived by the cells as abnormal condition, or because the hydrolytic enzymes actively promote these proteins activation? And if the second statement is correct, how it is possible that cathepsin are still active in the alkaline pH of the cell cytoplasm (even though slightly acidified by lysosomal content)?
Regarding the structure, the paper is quite well divided in paragraphs explaining the main drugs involved in lysosomal cell death. However, the separation of the drugs in the paper sections is a bit confusing: the authors talk about specific drugs (photosensitizers, immunosuppressant drugs, antihistamine drugs), grouping them according to their normal function as therapeutics. In other paragraphs the drugs mentioned are grouped according to their action on the lysosomes (lysosomotropic agents or lysosomotropic detergents) – I would suggest to put the more general part to the beginning of the article and the specific ones to the end. Specifically, chapter 3.4. and 3.5 should be placed before 3.3.
Noteworthy, there are several well investigated compound class, were the lysosome play an important part in their mode of action missing. Here, especially thiosemicarbazones like Dp44mT and DpC (Published by the Richardson group) as well as the clinically tested nintedanib need to be mentioned. Also the Kroemer Group and the Gibson Group published several articles which might be worth mentioning (e.g. PMID: 33414432, PMID: 36428570, PMID: 26859075). Thus, in total this article would profit from more profound literature research.
Author Response
Responses to Review 2
- In the introduction, reference is made to lysosomes and their importance in maintaining cellular homeostasis, with emphasis on autophagy as a specific therapeutic target, lysosome-dependent cell death and functional links between autophagy and apoptosis. Attention was paid to the possibility of regulating the permeability of the lysosomal membrane by numerous stressors, including drugs that have other clinical applications and which can be used in LMP induction, which may be important in anticancer therapy.
- Chapter 2 has been supplemented with information on the autophagy process. Differences between individual autophagic pathways, i.e. macro- and microautophagy and chaperone-mediated autophagy, were also clarified.
- Improved pH range in lysosomes.
- As suggested, chapter 5 was supplemented with additional groups of compounds affecting changes in the lysosomal system. Figure 2 and table 1 has also been supplemented with added LMP inducers.
- Changes were made in the order of the presented groups of compounds affecting lysosomes and subsections according to the Reviewer's suggestion.
- The scope of cited works has been increased.
- In subsection 4.1. an explanation of the action of cathepsins in the cytoplasm after their translocation from lysosomes has been supplemented.
The mechanism of action of cathepsins in the cytoplasm can be explained by acidification of the cytosol by hydrogen ions, which are released from the inside of lysosomes when their membranes are damaged (PMID: 16699952).
- Extensive corrections have been made to the English language in the manuscript.
We sincerely thank you for your time and review, which contributed to improving the quality of our manuscript.
Reviewer 3 Report
The reviewed manuscript Lysosomal path of death in cancer therapy seeks to summarize information about lysosomes as orchestrators of cell death and putative targets to be explored to stimulate cell death in malignant cells. The topic is obviously attractive and worth of exploration which is, unfortunately for this manuscript, reflected by a number of already published reviews focused on nearly identical or very similar theme. In reviewer´s opinion, these published papers provide most and in some aspects even more deeper information. It is felt that the reviewed manuscript does not bring someting substantial and mostly repeats what is already known. See for instance: Berg AL et al. . In: Mayrovitz HN 2022, Iulianna T et al.Cell Death Dis. 2022, Geisslinger F et al. Front Oncol. 2020 etc.
Author Response
Response to Review 3
We agree with the attention that the subject matter of our manuscript is covered in other works of the last few years. However, we believe that the issues presented in this paper, such as multidrug resistance, the search for new mechanisms of cell death and its inducers, are still very current problems. Therefore, in the manuscript we focused on the specific properties of lysosomes in cancer cells and on the phenomenon of lysosomal membrane permeability and its role in inducing cell death. We drew attention to groups of compounds that by their nature have clinical applications other than anti-cancer, but showing the property of affecting lysosomes, they can also be useful in oncological therapy (e.g. antihistamines). We emphasized the importance of combination therapy consisting in combining compounds from different groups and with different pharmacological properties in order to increase cytotoxicity to lysosomes of cancer cells, which should become one of the main targets of anticancer therapy.
Thank you very much for your time and review.
Reviewer 4 Report
Reviewers’ Comments (Remarks to the Authors):
The authors try to analyze from a current prism the mechanism by which lysosomes can modulate the process of tumor progression and the main derived therapeutic strategies.
In general, I reiterate that the manuscript is too general in terms of its content, without delving into the role of lysosomes in specific types of tumors or their escape pathways. Despite the novelty of the message, I have to say that the number of reviews on the role of Macroautophagy in cancer that have analyzed the role of these organelles has reduced my interest in some sections. Even so, I consider that the manuscript is suitable for publication.
Below I detail some issues to be changed or discussed by the team of authors. A detailed way and through a clear grammar should be improved by the authors in order encourage a reading by the reader, making the review attractive, being necessary to attract readers to the MDPI group.
Minor considerations :
1) The manuscript is well written, however as minor comment I recommend doing a revision of the English language to avoid the repetition of terms to describe the function of proteins, cell pathways or their expression levels, such as "increased" or "decreased", and use / combine more as "triggers" or "promotes". However, I reiterate that the grammatical level of the text is adequate
2) In general, the main text and the different sections are well referenced, however I recommend that the authors use more references that are between the 5-6 years immediately prior to writing the work (2021-22). A review text must know how to combine classic data, necessary to know the origin and evolution of the problem, together with current data that describe the current state of the topic.
3) I recommend authors rewrite the title of the manuscript, it is too general and repetitive in relation to the published bibliography. Place more emphasis on the idea of ​​using these organelles as therapeutic targets of great clinical potential.
Major considerations :
1) The abstract does not offer attractiveness to the reader, it must better focus the main idea of ​​the review. I recommend rewriting. Line 7-8: "case of many cancers (especially malignant tumor)"...all cancers are malignant tumors; authors should explain this summary better.
2) Introduction: Line 22 “the cancers diseases” is not well expressed. Lines 38-40: I don´t understand the message of this paragraph, authors must explain with other words or rewriting.
3) Section 3 Role of Lysosomes in cancer cells. this section does not delve into any particular idea. The authors should rewrite this section as it is one of the critical and hard points of the manuscript.
4) Sub-Section 3.1 Lysosomal Cell Death. Paragraph 106-111. Authors must explain the context and the main objective of this section. Subsection 3.2 and 3.3 is my opinion complement section 3.1. perhaps all of them should be grouped, facilitating the message and reading.
5) Section Plant Compound is not it is not numbered with respect to the other sections of the manuscript. what is the idea of ​​incorporating it? is it section 4 or 3.3.1….etcetera? Lines 163-174 is not clear the reading, I recommend read and re-explain the message.
6) Figure 2: what are the differences between Lysosomal and Autophagy cell death?. This key task have to integrated in the final format.
Author Response
Responses to review 4
- The manuscript has been extensively corrected in English.
- As suggested by the reviewer, the title of the manuscript was changed to: "Lysosomes as a target of anticancer therapy".
- The introduction was corrected, which emphasized the purposefulness of the type of cell death presented in the paper, which is lysosomal cell death, and the importance of LMP in anticancer therapy.
- Chapter 3 (Role of Lysosomes in Cancer Cells) has been revised to include factors contributing to the development of chemoresistance.
- In addition, Chapter 2 has been supplemented with information on the autophagy process. Differences between individual autophagic pathways, i.e. macro- and microautophagy and chaperone-mediated autophagy, were also clarified.
- Chapter 5 in the manuscript has been supplemented with additional groups of compounds affecting changes in the lysosomal system.
- The order of the groups of compounds acting on lysosomes described in the manuscript has been changed.
- In the introduction and in Chapter 1, the difference between autophagy and cell death based on lysosomal membrane permeability was highlighted.
- The scope of cited works has been increased in the manuscript.
We would like to thank you for preparing the review and valuable comments that contributed to improving the quality of our manuscript.
Round 2
Reviewer 2 Report
it is ok to publish the article now
Reviewer 3 Report
Authors have modified their work and refined some of the points which were originally felt as redundant or covered elsewhere. The manuscript is now more "readable", in particular due to the part dealing with lysosome-targeted therapies. Although much of what was criticized before still persists, i.e. general aspects of lysosomes and their activities are described in this manuscript and repeat what was already published before, it is felt that with editors persmission the current version of the manuscript could be published.
Reviewer 4 Report
Dear author
Thanks so much for your cooperation
In few days I will contact the editor in chief
Best Regards